# The piRNA cluster *torimochi* is an expanding transposon in cultured silkworm cells

**Keisuke Shoji**[1]*, **Yusuke Umemura**[2], **Susumu Katsuma**[3], **Yukihide Tomari**[1,4]*

**1** Laboratory of RNA Function, Institute for Quantitative Biosciences, The University of Tokyo, Bunkyo-ku, Tokyo, Japan, **2** Department of Life Sciences, Graduate School of Arts and Sciences, The University of Tokyo, Meguro-ku, Tokyo, Japan, **3** Department of Agricultural and Environmental Biology, Graduate School of Agricultural and Life Sciences, The University of Tokyo, Bunkyo-ku, Tokyo, Japan, **4** Department of Computational Biology and Medical Sciences, Graduate School of Frontier Sciences, The University of Tokyo, Bunkyo-ku, Tokyo, Japan

* kshoji@iqb.u-tokyo.ac.jp (K.S); tomari@iqb.u-tokyo.ac.jp (Y.T)

**Data Availability Statement:** The sequencing data obtained in this study are available under the accession number DRA014297 (https://ddbj.nig.ac.jp/resource/sra-submission/DRA014297), DRA015290 (https://ddbj.nig.ac.jp/resource/sra-

## Abstract

PIWI proteins and PIWI-interacting RNAs (piRNAs) play a central role in repressing transposable elements in animal germ cells. It is thought that piRNAs are mainly produced from discrete genomic loci named piRNA clusters, which often contain many "dead" transposon remnants from past invasions and have heterochromatic features. In the genome of silkworm ovary-derived cultured cells called BmN4, a well-established model for piRNA research, *torimochi* was previously annotated as a unique and specialized genomic region that can capture transgenes and produce new piRNAs bearing a trans-silencing activity. However, the sequence identity of *torimochi* has remained elusive. Here, we carefully characterized *torimochi* by utilizing the updated silkworm genome sequence and the long-read sequencer MinION. We found that *torimochi* is in fact a full-length gypsy-like LTR retrotransposon, which is exceptionally active and has massively expanded its copy number in BmN4 cells. Many copies of *torimochi* in BmN4 cells have features of open chromatin and the ability to produce piRNAs. Therefore, *torimochi* may represent a young, growing piRNA cluster, which is still "alive" and active in transposition yet capable of trapping other transposable elements to produce *de novo* piRNAs.

## Author summary

Transposons are DNA sequences that can jump from one region to another in the genome. Since transposon insertions can disrupt the genetic information, the hosts need to silence their activity especially in germ cells. PIWI-interacting RNAs (piRNAs) act as the sequence-specific guide to suppress transposons. Many piRNAs are generated from discrete genomic regions, called piRNA clusters, which often contain a variety of fragmented, "dead" transposons. piRNA clusters are thought to act as genomic storage devices that memorize non-self sequences to be suppressed. However, resent findings suggest that piRNA clusters are not stationary entities but can flexibly appear and disappear. BmN4 cells, derived from silkworm ovaries, have been widely used in the piRNA research. A

submission/DRA015290) (long DNA-seq). RNA sequences of differentiated BmN4 cells are available under the accession number DRA005335 (https://ddbj.nig.ac.jp/resource/sra-submission/DRA005335). Small RNA sequences of BmN4 cells (DRR181866), GFP repressed BmN4 cell line (DRR000974) and silkworm ovary (DRR086591), RNA sequences of BmN4 cells (DRR178519) and silkworm ovary (DRR186503), and ChIP sequences of IgG-R (DRR001873), H3K9me3 (DRR001878) and H3K27ac (DRR018643) antibodies were reported previously (Yokoi et al. 2021 (doi:10.3390/insects12060519); Izumi et al. 2020 (doi:10.1038/s41586-020-1966-9); Katsuma et al. 2021 (doi:10.1002/arch.21761), 2019 (doi:10.1002/2211-5463.12698); Kawaoka et al. 2012 (doi:10.1261/rna.029777.111), 2013 (doi:10.1093/nar/gks1275); Shoji et al. 2015 (doi:10.1093/nar/gku862)). The sequences of the transposons newly identified in this paper have been deposited in DDBJ under the following accession numbers: LC742510–5 (http://getentry.ddbj.nig.ac.jp/getentry/na/LC742510/ http://getentry.ddbj.nig.ac.jp/getentry/na/LC742511/ http://getentry.ddbj.nig.ac.jp/getentry/na/LC742512/ http://getentry.ddbj.nig.ac.jp/getentry/na/LC742513/ http://getentry.ddbj.nig.ac.jp/getentry/na/LC742514/ http://getentry.ddbj.nig.ac.jp/getentry/na/LC742515/). All code required for bioinformatics analysis in this paper is available at https://github.com/keishoji/torimochi.

**Funding:** This work is supported by MEXT | Japan Society for the Promotion of Science (JSPS) 18H05271 to YT; 16KT0064 to SK; 15J00469 to KS; 19K06484 to KS; 22K15082 to KS. The funders had no role in study design, data collection and analysis, decision to publish, or preparation of the manuscript.

**Competing interests:** The authors declare that they have no conflict of interest.

previous study using BmN4 cells identified *torimochi* (the Japanese word for birdlime) as a representative piRNA cluster, which can trap foreign sequences and produce piRNAs. However, the detailed properties of *torimochi* have remained unclear. Here, we found that *torimochi* is in fact a retrotransposon, which is exceptionally active and has massively expanded its copy number in BmN4 cells. Therefore, *torimochi* may represent a young, growing piRNA cluster, which is still "alive" and active in jumping yet capable of trapping other transposons to produce new piRNAs.

## Introduction

Transposons are DNA sequences that can migrate from one region to another in the genome. Since transposon insertions can disrupt the genetic structure, it is necessary for the hosts to suppress transposon activity especially in germ cells, where the genome is inherited to the next generation [1]. PIWI-interacting RNAs (piRNAs) and PIWI proteins play a central role in suppressing transposons in the germline [2, 3]. PIWI proteins use piRNAs as the sequence-specific guide to repress target transposons in two ways: transcriptional gene silencing (TGS) and post-transcriptional gene silencing (PTGS). piRNA-mediated TGS is achieved by heterochromatic histone modifications and DNA methylation [4, 5]. On the other hand, piRNA-mediated PTGS relies on the endoribonucleolytic activity of PIWI proteins [6–9]. The 3′ fragments of the cleavage products by piRNA-guided PIWI protein are incorporated into another PIWI protein and processed into new piRNAs [10, 11]. These new piRNAs further guide the cleavage of complementary RNAs (i.e., the same strand as the original piRNA), leading to the production of piRNAs with the same sequence as the original piRNAs. This cleavage-dependent piRNA biogenesis, called the "ping-pong" amplification cycle, is thought to be broadly conserved among many animals including flies, mice, zebrafish, sponges, and silkworms [5, 10, 12–14].

Most piRNA sequences are densely mapped to discrete genomic regions, called piRNA clusters [10]. The piRNA clusters often contain a variety of fragmented, "dead" transposon remnants deriving from past invasions, producing piRNAs that can recognize and repress currently active transposons [7, 9, 11, 15, 16]. In this way, piRNA clusters are thought to act as genomic storage devices that keep the information of non-self sequences to be suppressed.

The *flamenco* and *42AB* loci in *Drosophila melanogaster* represent the most well-studied piRNA clusters [10]. piRNAs derived from these piRNA clusters are produced in different ways: *flamenco* is a uni-strand piRNA cluster that produces Piwi-bound piRNAs through the phased piRNA biogenesis pathway, whereas *42AB* is a dual-strand piRNA cluster that produces Aubergine (Aub)/Ago3-bound piRNAs through the ping-pong piRNA biogenesis pathway [10, 11, 17]. Despite this difference, both piRNA clusters possess trimethylation of Lys9 on histone H3 (H3K9me3), a typical mark of heterochromatin [18]. These large heterochromatic piRNA clusters were originally proposed to provide the immunity to homologous transposon copies across the genome *in trans* [19, 20]. Indeed, *flamenco* is essential for transposon repression and oogenesis [21]. However, it was recently shown that *42AB* and some other dual-strand piRNA clusters are evolutionarily labile and mostly dispensable for transposon suppression, raising the possibility that dispersed elements in individual transposons can mediate silencing of themselves *in cis* [22]. Moreover, it has been reported that new euchromatic transposition of transposons can act as a source of piRNAs [23, 24]. Such euchromatic piRNA clusters are found not only in flies but also in mice and mosquitos [5, 23, 25]. These findings suggest that piRNA clusters are not stationary entities but can flexibly appear and disappear.

BmN4 cells, derived from silkworm ovaries, retain the active piRNA pathway including the ping-pong amplification cycle, and thus have been widely used in piRNA research [14]. Silkworms have no homolog of *Drosophila* Piwi, the PIWI protein that is specialized for the phased piRNA biogenesis pathway. Instead, silkworm Siwi participates in both the ping-pong and phased piRNA biogenesis pathways [26]. A previous study using BmN4 cells identified *torimochi* (the Japanese word for birdlime) as a piRNA cluster in silkworms [27]. *Torimochi* was regarded as a unique and specialized piRNA cluster in the BmN4 genome, which traps foreign transgenes and produces piRNAs that can repress homologous transgenes in other genomic regions *in trans* [27]. However, the detailed sequence properties of *torimochi* have remained unclear.

The previous *torimochi* study was conducted using the old silkworm genome sequence originally published in 2008 [27, 28]. At that time, the silkworm genome contained many unassembled regions, even within *torimochi*. Accordingly, there was only one genomic region to which *torimochi*-derived piRNAs were mappable, and the apparent difference in the chromosomal position of *torimochi* between BmN4 cells and silkworm ovaries was thought to be due to a large rearrangement of the corresponding genomic region [27]. However, the silkworm genome sequence has recently been updated with completing most of the previously unassembled regions [29]. In this study, we carefully reexamined the *torimochi* region to understand why it can trap foreign transgenes for silencing. Using the updated silkworm genome sequence and the long-read sequencer MinION, we found that *torimochi* is in fact a full-length gypsy-type transposon, which has massively expanded its copy number in BmN4 cells compared to silkworms. Moreover, our single-nucleotide polymorphism (SNP) and chromatin immunoprecipitation (ChIP) analyses indicated that many copies of *torimochi* in BmN4 have features of open chromatin and the ability to produce piRNAs. Therefore, *torimochi* may serve as a model for young piRNA clusters, which are still "alive" and active in transposition, can trap other transposons, and produce *de novo* piRNAs.

## Results and discussion

### The piRNA cluster *torimochi* is a full-length gypsy-type transposon

In the original silkworm genome sequence published in 2008, *torimochi* was annotated as a specialized piRNA cluster located on chromosome 11, because no highly homologous region was found elsewhere in the genome [27, 28]. Moreover, the chromosome 11 *torimochi* itself contained a large unassembled region. However, recent improvements in sequencing technology have determined the sequences of many previously unassembled regions in the silkworm genome [29]. Using the updated silkworm genome sequence published in 2019, we performed a nucleotide BLAST search using the known *torimochi* sequence as a query. We found that there are at least three copies of *torimochi*-like sequences, not only on chromosome 11 but also on chromosomes 12 and 24 (Fig 1A). Alignment of these three copies of *torimochi*-like sequences suggested that they are full-length gypsy-type retrotransposons, which even retain the conserved LTR sequences at both ends (Fig 1B). Therefore, *torimochi* is not a unique sequence in the genome as was thought in the past, but should be now interpreted as a gypsy-type transposon with multiple copies in the genome.

We found that *torimochi*-derived piRNAs are highly expressed in BmN4 cells but not in silkworm ovaries, suggesting that *torimochi* has gained the piRNA-producing ability in BmN4 cells (Fig 1C). Most (~73%) of the *torimochi*-derived small RNAs were ~26–32 nucleotides long (S1A Fig), which showed both the ping-pong signature (S1B Fig) and the head-to-tail phasing signature (S1C Fig). Therefore, they can be regarded as bona fide piRNAs. To further characterize the piRNA-generating activity of *torimochi*, we compared the patterns of piRNA

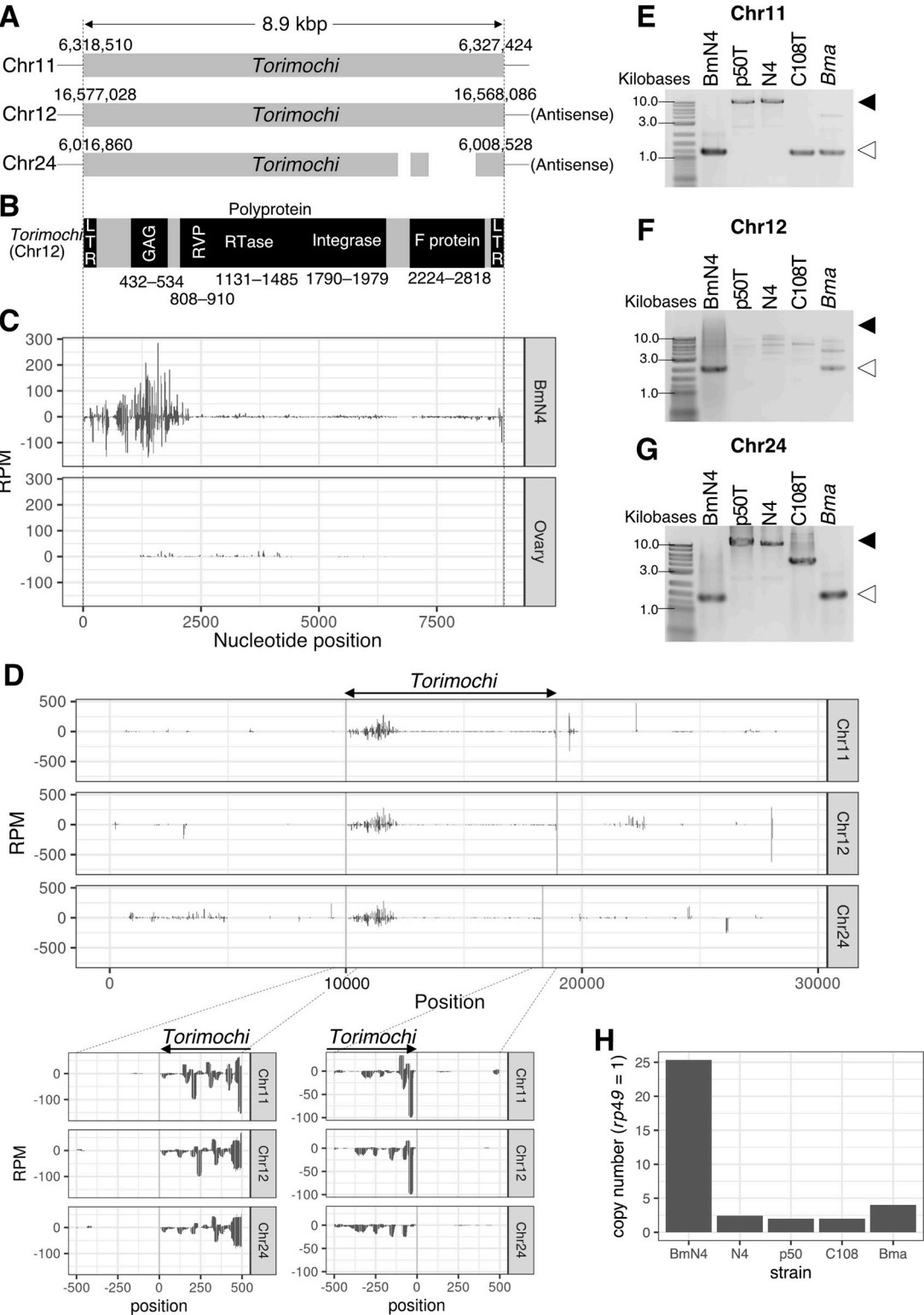

**Fig 1. _torimochi_ is a full-length, gypsy-type transposon.** (A) Schematic representation of the three copies of _torimochi_ in the recently published silkworm genome. The copy on chromosome 24 has a deletion in the region encoding F protein. (B) Structure of _torimochi_ on chromosome 12. There are LTR sequences at both ends, with retrotransposon gag protein (GAG), retroviral aspartyl protease (RVP), reverse transcriptase (RTase), integrase, and F protein (Baculovirus F protein) encoded inside. RVP, RTase, and integrase are encoded as polyproteins without stop codons. Black boxes show the reading frame and the numbers

below indicate the amino acid numbers. (C) Distribution of *torimochi*-derived piRNAs in BmN4 cells and silkworm ovaries. Although *torimochi* on chromosome 11 is used as a representative, piRNAs produced from *torimochi* on other chromosomes can be mapped in the same way due to sequence similarity. The positive and negative values on the y-axis indicate the coverage of piRNAs in the sense and antisense directions, respectively (the same hereafter). (D) Distribution of piRNAs derived from the regions inside and outside of *torimochi* in the recently published silkworm genome. (E–G) Genomic PCR using primers outside of *torimochi* on chromosome 11 (E), chromosome 12 (F), and chromosome 24 (G). The genome DNAs of BmN4 cells and silkworm strains p50T, N4, and C108T and *B. mandarina* (*Bma*) were used. Black, white arrows indicate the band lengths with and without *torimochi*, respectively. The PCR amplification efficiency of the full-length *torimochi* on chromosome 12 was extremely low due to its sequence context (see S1E Fig). (H) Estimation of the copy number of *torimochi* by qPCR of genomic DNA. The copy number of *rp49* (a gene on an autosomal chromosome) was normalized to 1.

production inside and outside of *torimochi* for the three insertion sites on the BmN4 genome. Every insertion site showed clear boundaries for piRNA production; piRNAs were densely mapped within the *torimochi* region, whereas the outside region was almost devoid of piRNAs (Fig 1D). This observation suggests that *torimochi* is a full-length transposon that acts as a piRNA-producing cassette.

The 2019 silkworm genome was based on the p50T strain of silkworm [29]. Given that *torimochi* is a full-length transposon that could in theory retain transposition activity, the actual locations of *torimochi* may be different between the genomes of P50T strain and BmN4 cells. Therefore, we performed quantitative genomic PCR of three silkworm strains (p50T, N4, and C108T), *Bombyx mandarina* (*Bma*, a wild progenitor of *Bombyx mori*), and BmN4 cells. We found that the *torimochi* insertion status at the above-identified three genomic locations greatly varies among strains. In fact, the BmN4 genome lacks *torimochi* insertion at any of the three above-mentioned locations (Fig 1E–1G). Nevertheless, when we measured the copy number of *torimochi* by qPCR, we found that *torimochi* in BmN4 cells has ~25-fold more copies than the r*ibosomal protein 49* (*rp49*) gene on the autosomes, which is remarkably higher than those in the *B. mori* strains and *B. mandarina* (Fig 1H). These results suggest that *torimochi* is an active transposon unit that keeps transpositioning in the silkworm genome and that BmN4 cells have gained massive copy numbers of *torimochi* presumably during the establishment of the cell line and/or cell passage.

## Detailed annotation of *torimochi* in the BmN4 genome by utilizing MinION sequencer

To investigate where *torimochi* copies are inserted in the BmN4 genome, we opted to use the long-read sequencer MinION. We processed the MinION reads to search for the junctions between the known *torimochi* sequences and other genomic regions in BmN4 cells (Fig 2A). Among them, we identified at least nine sites that have reliable junctions at both ends and with a reasonable genomic distance. Genomic PCR showed two distinct lengths of bands—with and without *torimochi*—for eight of those nine sites, indicating that *torimochi* is inserted in a heterozygous manner at each site (S2 Fig). For the insertion in chromosome 22, bands of incomplete lengths were observed, so we decided to exclude it from further analyses for simplicity (S2 Fig). We then obtained the precise sequences of these eight *torimochi* insertions by Sanger sequencing of PCR-amplified fragments and constructed a phylogenetic tree together with the above-identified three *torimochi* insertion sequences in the p50T silkworm genome. We found that all the eight *torimochi* sequences in the BmN4 genome are similar to each other, belonging to one clade that is distant from the three *torimochi* copies in p50T strain (Fig 2B). This suggests that expansion of *torimochi* in BmN4 cells has occurred relatively recently. In the eight homologous *torimochi* sequences in the BmN4 genome, we could identify 61 SNP positions that can help distinguish them (Fig 2C). At each position of these SNPs, we analyzed the nucleotide composition of BmN4 piRNAs using previously published data [26].

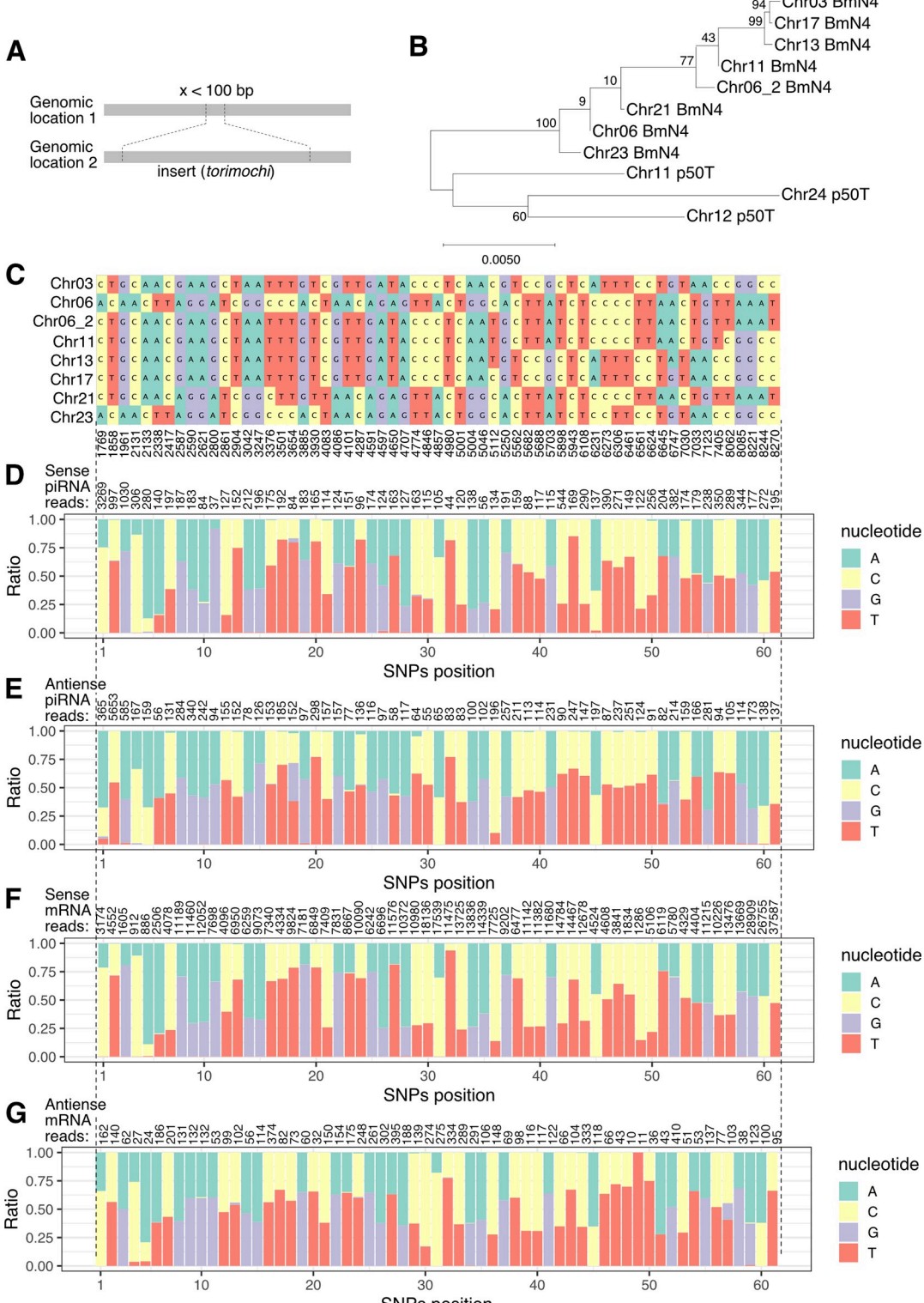

**Fig 2. Detection of novel *torimochi* inserts in the BmN4 genome.** (A) Definition of inserts in this paper. When the junctions between the two genomic locations are less than 100 bp apart at the genomic location 1 and are consistent in orientation, the region between the junctions at the genomic location 2 is considered as an insert. (B) Phylogenetic tree of the new *torimochi* insert sequences in BmN4 cells and the *torimochi* copies present in p50T strain. Numbers indicate bootstrap probabilities. (C) List of SNPs in each *torimochi* in BmN4 cell. The numbers below indicate the position of the SNPs in *torimochi* on

chromosome 3. (D–G) Distribution of SNPs at each position in sense (D, F) and antisense (E, G) piRNAs (D, E) and mRNAs (F, G) deriving from the new *torimochi* inserts in BmN4 cells. SNPs position indicates the order of the SNPs shown in (C). The read coverages of the SNPs site are shown above the graphs.

We found piRNA and mRNA sequence polymorphisms at all the SNP positions (Fig 2D–2G), suggesting that *torimochi* piRNAs and mRNAs are produced from many different *torimochi* loci, rather than a single locus. Comparison of the piRNA production patterns inside and outside of *torimochi* for these eight insertion sites showed clear boundaries for piRNA production (S3A–S3H Fig). These observations support the model that *torimochi* is a full-length, multi-copy transposon, each copy of which acts as a piRNA-producing cassette.

In the previous *torimochi* paper, a GFP transgene, introduced via the piggyBac transposon, was trapped by *torimochi* and silenced by *de novo* piRNAs in BmN4 cells [27]. To identify which *torimochi* copy newly annotated in the BmN4 genome the GFP transgene was actually inserted into, we analyzed the genome of #8 cells, which produce abundant GFP-derived piR-NAs [27], using MinION. We found two reads that cover the GFP cassette, *torimochi*, and the neighboring silkworm genome at the 11,929 kb position on chromosome 13 (S3I Fig). Mapping of piRNAs showed that piRNA are produced mainly within the *torimochi* cassette rather than the surrounding regions (S3J Fig), just like other *torimochi* copies on different chromosomes (S3A–S3H Fig). These data suggest that, unlike the original assumption that *torimochi* is a unique, specialized piRNA cluster, the *torimochi* unit that captured the GFP transgene in #8 cells is merely one of many *torimochi* copies in the BmN4 genome.

## Comprehensive identification of transposons activated in the BmN4 genome

Given that *torimochi* turned out to be an active, multi-copy transposon, we decided to extend our analysis to other transposons in silkworms. We searched for novel transpositions in the BmN4 genome in an unbiased manner, using MinION sequencing and the same criteria as for *torimochi* (Fig 2A). In this analysis, we defined "novel transpositions" when those sequences that existed in the p50T genome (published in 2019) are found elsewhere in the BmN4 genome. We identified ~700 new inserts that have reliable junctions at both ends. The estimated size of these inserts was mostly in the range of $10^2$ to $10^4$ base pairs, so we decided to focus on the 597 inserts that fall within this range (Fig 3A). These predicted inserts can be either full-length transposons, fragments of transposons, or transposon-unrelated sequences. Therefore, we categorized them into groups by sequence homology; when locally homologous sequences were bridged by a longer insert(s) (e.g., sequence 4), we considered them all as one group (Fig 3B). Then, we further focused on 20 groups that include at least five inserts (Fig 3C). Of these, remarkably, Group 2 that had the second most insert sites represented *tori-mochi* (and its fragments), while Group 1 with the highest number of inserts turned out to represent LINE transposon (and fragments). For each group, the longest sequence was selected as the representative and annotated by nucleotide BLAST with the *B. mori* transposon database (S2 Table). As a result, they were identified to be LINE or SINE (Groups 1, 7, 9, 11, 15), known transposons (Groups 5, 6, 10, 12, 13, 16, 18, 19), *torimochi* (Group 2), or novel sequences (Groups 3, 4, 8, 14, 17, 20) (S2 Table). We found that LTR transposons like *torimochi* and transposons with Inverted Terminal Repeat (ITR) tend to retain their full-length copies upon new transposition (Figs 3D, S4E, S4G, S4H, S4I, S4K and S4L), whereas non-LTR transposons have produced incomplete copies fragmented from the 5' side (S4A–S4C Fig). These results suggest that the new insert sequences identified in this study are "alive" and active transposons that have transposed relatively recently. For these novel transposons, we named group 3 as

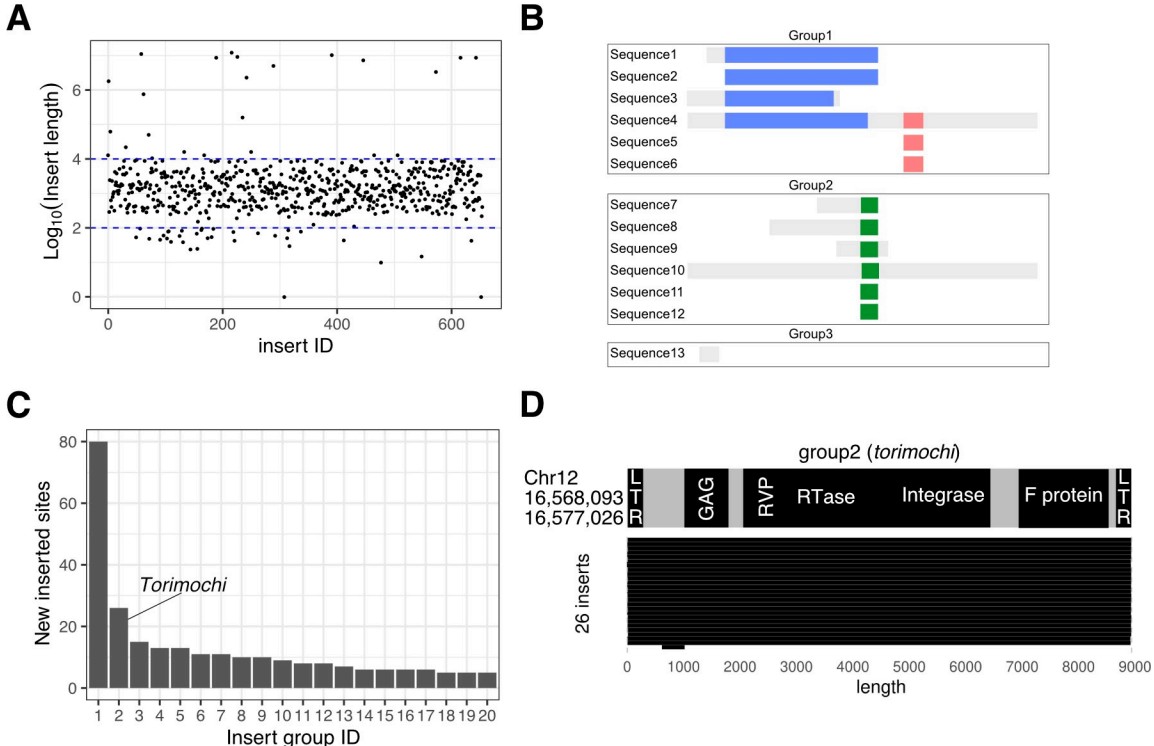

**Fig 3. Comprehensive identification and grouping of novel insertion sequences.** (A) Estimated lengths of novel inserts in the BmN4 genome. Blue dotted lines indicate 100 bp and 10,000 bp. (B) Schematic diagram of grouping. Segments with the same color are homologous to each other. For example, sequences 1 and 6 do not have direct homology but are bridged by sequence 4, thereby categorized into the same group. (C) The numbers of inserts in each group. Those groups that had 5 or more insert sites were sorted. Group 2 represents *torimochi*. (D) Domain structure of the representative sequence of group2 (*torimochi*) (top) and the region where it is originally annotated in the p50T genome (left). The lengths of the inserts found in BmN4 cells are shown at the bottom.

*mejiro* (Japanese white-eye, which is often trapped by birdlime [torimochi]; see S5 Fig), group 4 as *TRAS-lbm* (***TRAS-l**ike in **Bm**N4*; homologous to *TRAS* transposon), non-autonomous (transposase-lacking) transposons of group 8 as *kotaro*, 14 as *kojiro*, 17 as *kotetsu* (popular Japanese cat names; cats jump up and down, attracted by catnip [transposase from another autonomous transposon]), and group 20 *wao* (**w**andering p**ao**; homologous to *pao* transposon), respectively (S4A, S4B, S4E, S4G, S4I and S4L Fig).

## *torimochi* is the most specifically activated transposon in BmN4 cells

Next, we compared the published genome sequence of the p50T strain and our MinION-based genome sequence of BmN4 cells. We found that p50T already has >70,000 copies of Group 1 (LINE-associated) insertions but only 10 copies of Group 2 insertions (including three full-length *torimochi* and seven partial *torimochi*; S3 Table). To quantitatively evaluate the expansion of insertions in BmN4 cells, we calculated the ratio of the total genomic areas between the "old" copies already existing in the p50T genome and the "new" copies specifically found in the BmN4 genome for each insertion group. Notably, the ratio was the highest for *torimochi*, indicating that *torimochi* is the most massively expanded insertion in BmN4 cells (Fig 4A). At the same time, we found that not only *torimochi* but also many other insertions (Groups 3, 8, 17, 4, 19 and 20) have increased their copies in BmN4 cells, most of which correspond to the novel transposons identified in this study (Fig 4A).

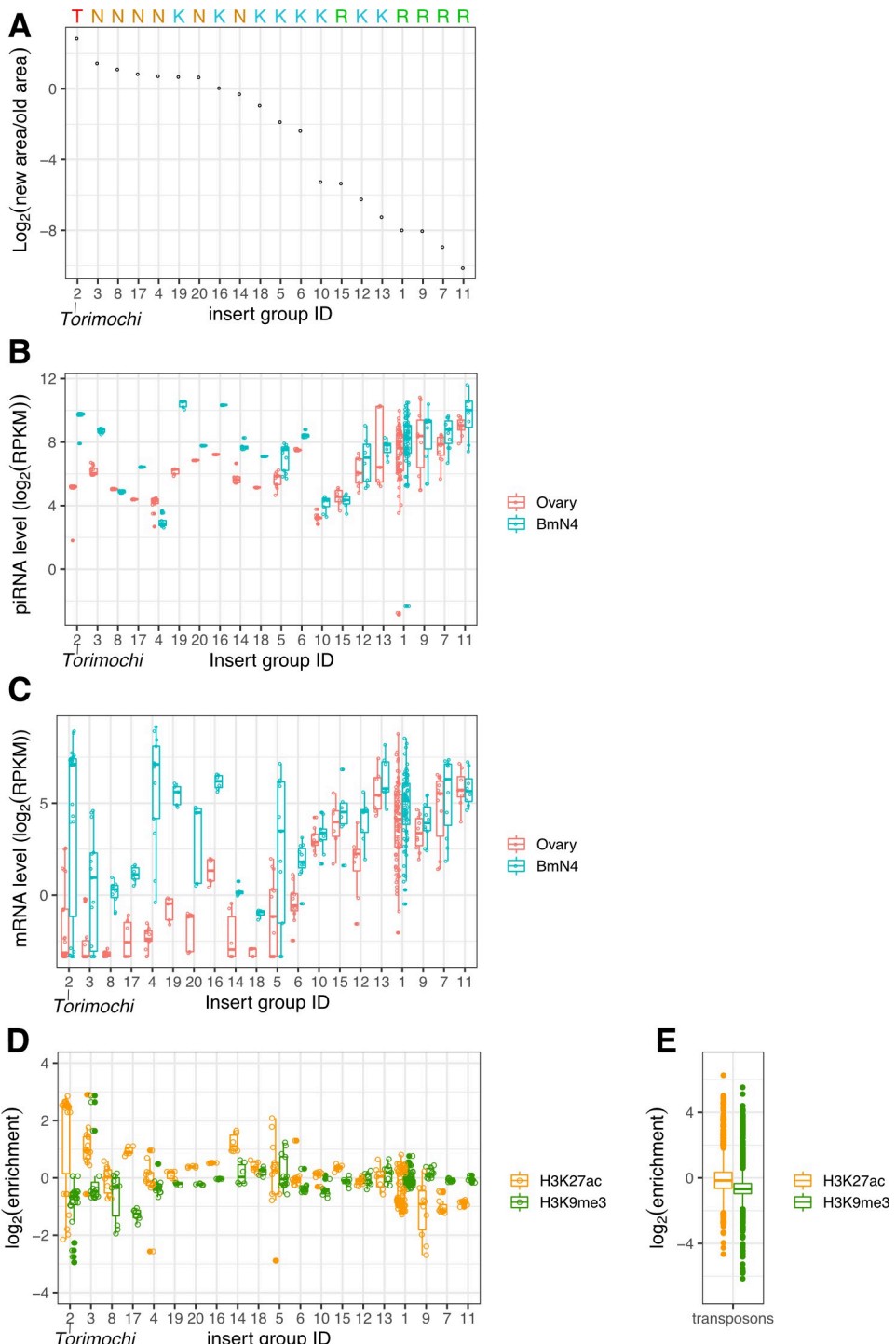

**Fig 4. *torimochi* is the most active transposon in BmN4 cells.** (A) Ratios between "new" areas in BmN4 cells and "old" areas in p50T strain for the inserts in each group. *torimochi* (group 2) is the most highly activated insert group in BmN4 cells. T (*torimochi*), N (novel transposons), K (known transposon), R (SINE/LINE) are shown (top). (B) The amounts of piRNAs derived from the inserts in each group in ovaries and BmN4 cells. The order is the same as in (A). Active inserts tend to produce more piRNAs in BmN4 cells than in ovaries. (C) The amounts of mRNAs derived from the inserts in each group in ovaries and BmN4 cells. The order is the same as in (A). Active inserts tend to produce more mRNAs in BmN4 cells than in ovaries. (D) Enrichment of the H3K27ac euchromatin mark and the H3K9me3 heterochromatin mark compared to IgG (control) of the inserts in each group BmN4 cells. The order is the same as in (A). (E) Enrichment of the H3K27ac euchromatin mark and the H3K9me3 heterochromatin mark for 1811 known transposon sequences.

To investigate if these BmN4-specific new insertions have the ability to produce piRNAs like *torimochi*, we compared piRNA production from these insertions in silkworm ovaries and BmN4 cells (Fig 4B). We found that most of the "new" insertions (left) show significantly increased piRNA production specifically in BmN4 cells, whereas the piRNA levels from "old" insertions such as LINE transposon (right) were comparable between silkworm ovaries and BmN4 cells (Fig 4B). This increase of piRNA levels correlated well with the increase of mRNA levels in BmN4 cells (Fig 4C). Previous ChIP-seq analysis revealed that *torimochi* is a transcriptionally active region with high levels of euchromatic modifications [30]. We therefore investigated the degree of histone modifications for these "new" inserts. Remarkably, these "new" insertions with piRNA-producing ability showed enrichment of the euchromatin mark H3K27ac and the depletion of the heterochromatin mark H3K9me3 (Fig 4D), compared to their average levels in all sequences in the silkworm transposon database (Fig 4E). These results suggest that these "new" insertions have an open chromatin structure, as previously reported for *torimochi* [27, 30]. Nevertheless, it is noteworthy that, among those "new" insertions, *torimochi* has the highest activity in transposition and the most open chromatin structure (Fig 4A and 4D), which may explain why the GFP transgene was trapped by *torimochi* for piRNA production in the previous study [27]. This is not limited to the GFP transgene; we identified as many as 36 insertions of transposons with at least one end forming a well-defined junction with *torimochi* (S5A Fig). Notably, we found that the Group 3 transposon *mejiro*, the second most active transposon in BmN4 cells, is inserted into seven different sites within *torimochi* (S5A Fig), one of which was confirmed by PCR to be BmN4-specific (S5B and S5C Fig.). Thus, *torimochi* is active in trapping not only exogenously introduced transgenes but also endogenous transposons.

To determine when *torimochi* became active, we compared *torimochi* insertion sites using our standard line of BmN4 cells (originally from Kyushu University; referred to as BmQ here) and another line of BmN4 cells that had been maintained in Yamaguchi University for decades and then have maintained in our laboratory for ~10 years [31] (referred to as BmY here). Initial screening of the *torimochi* insertion sites in the genome of each BmN4 line identified 26 sites in BmQ and 17 sites in BmY, only two of which appeared to be common in both lines. However, it was possible that our standard criteria for defining the *torimochi* insertion sites were too stringent. Therefore, we directly mapped the reads of the MinION sequence at each identified insertion site and calculated the ratio of reads with or without *torimochi* insertion (S6A Fig; note that the BmN4 genome is highly polyploidized, so each chromosomal copy may or may not contain the *torimochi* sequence at each defined insertion site). As a result, all the 26 insertion sites defined in the BmQ genome turned out to contain the *torimochi* insertion sequence even in the BmY genome at similar presence/absence ratios, and vice versa for 16 out of the 17 insertion sites defined in the BmY genome (S6B Fig). There was one insertion site that was present in the BmY genome but not in the BmQ genome. The results suggest that the majority of *torimochi* transpositions had occurred more than a few decades ago, most likely upon establishment of the original BmN4 cell line, and yet new insertion events are still ongoing.

## *torimochi* is transcribed by the BmN4 cell-specific transcription system

It is known that well-established piRNA clusters in *Drosophila* (e.g., the *42AB* cluster) have a specialized system for transcriptional activation. However, those specialized transcriptional activators such as the Rhino-Deadlock-Cutoff (RDC) are conserved only within the *Drosophila* genus, and thus the transcriptional activation systems of piRNA clusters are likely to be different in different organisms [32–34]. Keeping this in mind, we asked if the transcription

mechanism of *torimochi* is any different from other piRNA-producing transposons in BmN4 cells. Although specific transcriptional activators of piRNAs clusters remain unknown in silkworms (as in many other animals except for *Drosophila*), there is a method to differentiate BmN4 cells into adipocyte-like cells as performed in mammalian 3T3-L1 cells [35]. Therefore, we decided to differentiate BmN4 cells into adipocytes so that they lose their "germline-ness" [35]. As expected, the expression of the adipocyte marker *BmFABP1* (Fatty Acid-Binding Protein 1) was markedly increased (Fig 5A), while the expression levels of piRNA-related factors such as *Vasa* were decreased (Fig 5B). Importantly, transcription of *torimochi* was drastically reduced by adipocyte differentiation (Fig 5C), whereas most other transposons, including those piRNA-producing transposons in BmN4 cells, remained unrepressed or rather increased by differentiation (Fig 5C and 5D). These findings suggest that, even among those piRNA-producing transposons in BmN4 cells, *torimochi* has started to gain a specialized, germline-specific transcriptional activation system.

In this study, we found that *torimochi* is not a specialized piRNA cluster but in fact a full-length gypsy-type transposon that is exceptionally active in BmN4 cells. It was recently reported that large piRNA clusters are evolutionarily labile and can be deleted without compromising transposon regulation in *Drosophila* [22]. Thus, piRNA clusters in the genome are not as static as previously thought but can flexibly appear and disappear. Indeed, *torimochi* does not serve as a source of piRNAs in silkworm ovaries but has massively expanded its copy number and gained the activity to produce piRNAs in BmN4 cells (Fig 1). *Torimochi* has the open chromatin structure and can trap foreign transgenes as well as endogenous transposons (Figs 4, S3I and S5). Moreover, our SNP analysis showed that piRNAs are produced from many different copies of *torimochi* in the BmN4 genome. Notably, unlike other transposons, *torimochi* seems to be integrated into a specialized, BmN4-specific transcription system (Fig 5). Therefore, *torimochi* in BmN4 cells may represent a young, growing piRNA cluster, which is still "alive" and active in transposition but capable of trapping other transposable elements to produce *de novo* piRNAs. Moreover, we successfully identified six novel transposons that have expanded their copies and gained the piRNA-producing activity specifically in BmN4 cells, just like *torimochi* (S4 Fig). Future studies should focus on how transposons gain the ability of piRNA production and what hallmark features active piRNA clusters have.

## Materials and methods

### Cell lines

BmN4 cells (BmQ provided by Chisa Yasunaga-Aoki, Kyushu University, and BmY provided from Dr. Jun Kobayashi, Yamaguchi University) were cultured at 27˚C in IPL-41 medium (Applichem) and TC-100 medium (Applichem) supplemented with 10% fetal bovine serum, respectively.

### Search for *torimochi* sequences in the genome and their annotation

*torimochi* sequences were BLAST-searched in SilkBase (http://silkbase.ab.a.u-tokyo.ac.jp/) using the sequence information described in a previous paper [27] as a query in the updated silkworm genome [29]. Since there were homologous regions on multiple chromosomes, the sequences were extended to the 5′ and 3′ ends, and then the common regions and Long Terminal Repeat (LTR) regions were identified by BLAST. The obtained nucleotide sequences were analyzed by pfam to estimate the domain structure of the encoded proteins (http://pfam.xfam.org/).

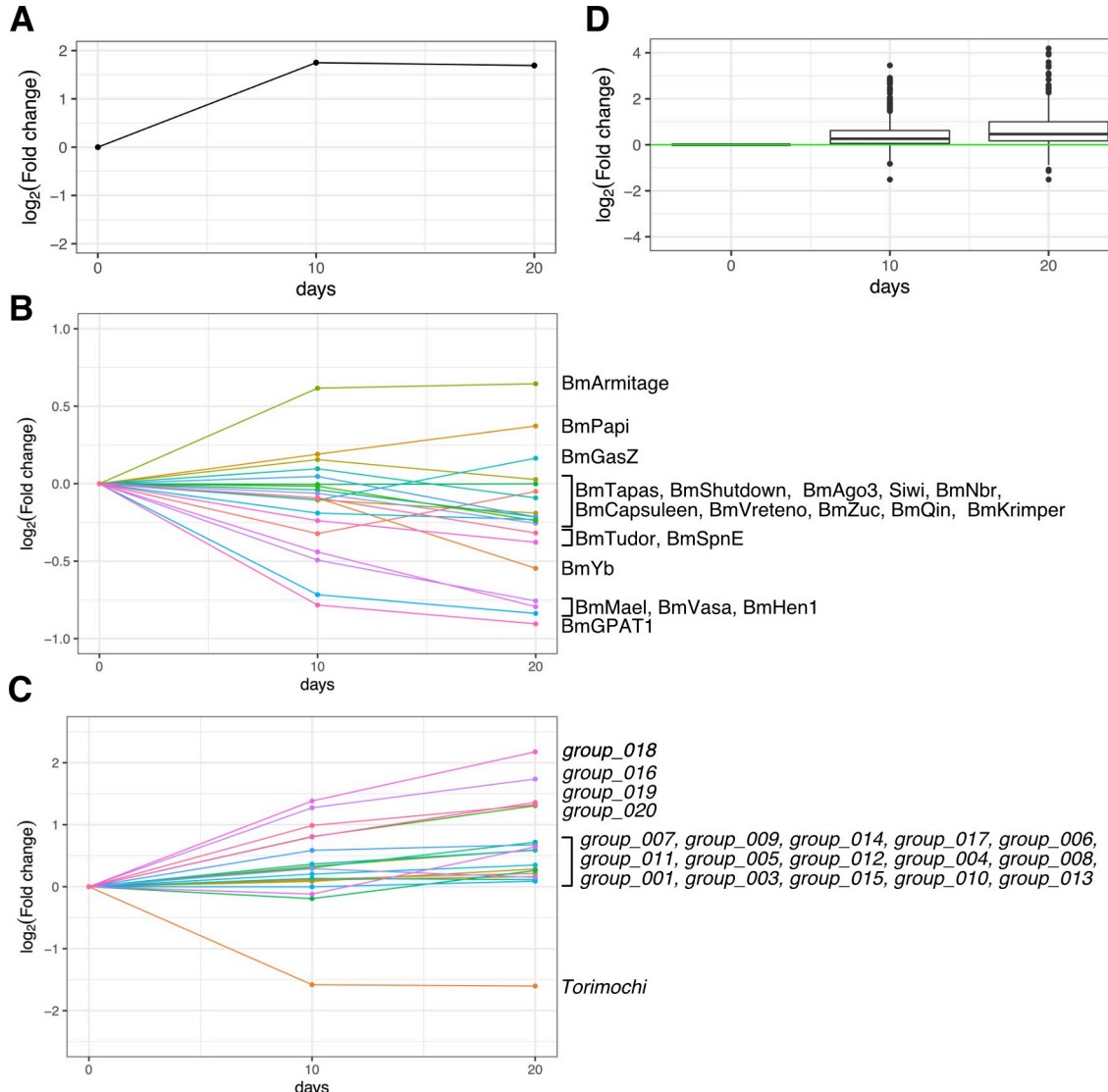

**Fig 5. Downregulation of *torimochi* by differentiation of BmN4 cells into adipocytes.** (A) Relative expression levels of *BmFABP1* (Fatty Acid-Binding Protein 1) during differentiation of BmN4 cells into adipocytes, calculated from the RNA-seq data. The expression levels at day 0 (untreated) were used as the baseline. *BmFABP1* is upregulated by differentiation. (B) Relative expression levels of piRNA-related genes during differentiation of BmN4 cells into adipocytes. The crowded parts are labeled in the order of the relative expression levels at day 20. The expression levels at day 0 (untreated) were used as the baseline. Many piRNA-related factors are downregulated by differentiation. (C) Relative expression levels of the insert groups identified in Fig 4 during differentiation of BmN4 cells into adipocytes. The crowded parts are labeled in the order of the relative expression levels at day 20. The expression levels at day 0 (untreated) were used as the baseline. *torimochi* is downregulated by differentiation, whereas other transposons are upregulated. (D) Box plots of the relative expression levels of transposons during adipocyte differentiation. 579 transposons with the RPKM values greater than 10 in any of the libraries on day 0, 10, or 20 were plotted. Center line, median; box limits, upper and lower quartiles; whiskers, 1.5 × interquartile range; points, outliers.

## Sequence analysis of deposited small RNAs for known *torimochi*

Small RNA sequence of BmN4 cells (DRR181866) and silkworm ovaries (DRR086591) were reported previously [26, 36]. Informatics analysis of small RNAs was performed as reported previously [37]. In brief, the 3′-adaptor sequences were identified and removed by cutadapt 2.6 with Python 3.7.0 with "—minimum-length 20" parameter [38] Reads that could be

mapped by bowtie 1.2.3 [39] to the *Bombyx mori* genome [29] up to two mismatches were used to calculate the mapping rate and to normalize each library. Sam files were converted to bam files by SAMtools [40], then to bed files, and the coverage of each nucleotide was calculated by BEDTools [41]. The *torimochi* sequence on chromosome 11 was used as a representative of *torimochi* sequences. The 10 kbs regions before and after the sequences homologues to the chromosome 11 *torimochi* sequence were extracted using BEDTools fastaFromBed [41]. Two nucleotide mismatches and multi-maps were allowed when mapping small RNAs to those sequences using bowtie [39]. Of the mapped results, only small RNAs of ≥23 nucleotides were extracted and treated as piRNAs, and the coverage of each nucleotide was calculated using the "-d, -s, -S" options of coverageBed in bedtools. For the calculation of the ping-pong signature, *torimochi*-mapped small RNAs (23–32 nucleotides) were extracted and the frequency of overlap at the 5′ end was calculated. The z-score was calculated by dividing the difference from the mean scores of the background by the standard deviation of the scores the background, defined as distances of 1–9 and 11–20 nucleotides. To calculate the phased signature, *torimochi*-mapped small RNAs of ≥23 nucleotides were extracted and the frequency of the distance between the 3′ end and the downstream 5′ end was calculated. The z-score was calculated by dividing the difference from the mean scores of the background by the standard deviation of the scores of the background, defined as distances of 0 and 2–19 nucleotides. Results were imported into R and graphed.

### Genome extraction and genomic PCR

Genomic DNA was extracted from BmN4 cells using the Blood and Tissue kit (QIAGEN) according to the protocol of the manufacturer. Genomic DNA from *Bombyx mori* and *Bombyx mandarina* was provided by Takashi Kiuchi. PCR was performed by step-down PCR using KOD One (TOYOBO), and the PCR products were analyzed in 1% agarose with 1 kb DNA Ladder (New England Biolabs). Primers used in this study are shown in S1 Table.

### Quantitative PCR for measuring the copy number of *torimochi*

Genomic DNA was used as a template for quantitative PCR analysis on a Thermal Cycler Dice Real Time System (TaKaRa) using KAPA SYBR Fast qPCR Kit (KAPA Biosystems) and the primers shown in S1 Table Absolute quantification of the copy number was performed using a known amount of plasmid containing the primer region. The primer set corresponding to the *rp49* gene on the autosome was used as a control.

### MinION-seq and mutation detection

Genomic DNA was extracted from BmN4 cells using Blood & Cell Culture DNA Kit (QIAGEN), sequencing libraries were prepared using Rapid Sequencing Kit (Oxford Nanopore), and the data were acquired by MinION using flow cell (version 9). The obtained sequences were mapped to the silkworm genome by ngmlr, and the variants of genomic structure were detected by svim with "–nanopore" option [42, 43]. Then, the structural variants of BND (breakends; connection of distal genomic positions) were extracted, and the 597 inserts that have reliable junctions at both ends were obtained by R script. Blastn was performed using the silkworm genome, known transposon sequences, and the 597 new insert sequences. Groups were created by combining sequences with e.values of less than 1e-100 among the new inserts (Fig 3B). Then, for each group, the regions in the silkworm genome that showed e.values of less than 1e-100 were identified as the regions of homology. The longest insert sequence in each group was used as the representative sequence and its sequence annotation was performed by pfam (http://pfam.xfam.org/). For novel transposon sequences that did not show

homology to known transposon sequences, Sanger sequencing was performed, and the determined sequences were registered in the repository (the accession numbers: LC742510–5).

### Cloning of *torimochi* and phylogenetic tree analysis

The sequences outside each *torimochi* copy were used to design the primers to distinguish it from other copies. PCR was performed by using inside-outside and outside-inside primer sets for each *torimochi* copy, and the PCR products were purified using NucleoSpin Gel and PCR Clean-up (MACHEREY-NAGEL) and then sequenced. *torimochi* sequences were clustered by Clustal W in MEGA X [44] and the phylogenetic trees were generated by the maximum likelihood method with bootstrapping.

### SNPs detection

The sequence of each *torimochi* copy was combined and aligned with MEGA X [44], and SNP-sites [45] was used to generate a list of polymorphisms. The piRNA sequence library and the RNA sequence library divided into 33 nucleotides each [26] were mapped to the *torimochi* sequence on chromosome 3 using bowtie, allowing two mismatches and multi-mapping [39]. Sam files were converted to bam files by SAMtools [40], and the SNP information of piRNAs was extracted using a custom R script.

### Detection of EGFP transgene inserted *torimochi*

The genomic sequence of the GFP repressed BmN4 line [27] was obtained by minION and analyzed by svim [43], and the IDs of the reads mapped to both GFP and *torimochi* were extracted by using the information in "candidates_breaksends.bed" of svim output. The region on the genome corresponding to the extracted reads were identified using BLAST on SilkBase (http://silkbase.ab.a.u-tokyo.ac.jp/), and the sequence was reconstructed for the following analysis. The piRNA sequences of the GFP repressed line were mapped to the reconstructed sequence using bowtie, allowing one mismatch and multi-mapping. Of the mapped reads, only those with ≥23 nucleotides were used to calculate the coverage of each nuelcotide using coverageBed and the "-d, -s, -S" options [41].

### Mapping of ChIP-seq, mRNA-seq, and piRNA-seq to novel inserts and transposons

Previously published libraries of ChIP-seq, mRNA-seq and piRNA-seq [26, 27, 30, 36, 46–48] were mapped to the newly identified inserts using hisat2 (ChIP-seq, mRNA-seq) and bowtie (piRNA-seq), and the numbers of length-corrected mapped reads were calculated [39, 49]. Sam files were converted to bam files by SAMtools [40], then to bed files, and the coverage of each nucleotide was calculated by BEDTools [41].

### Detection of transposition of transposon inside *torimochi*

The BmN4 genome sequences were mapped to the known transposons and newly identified transposons in this paper by ngmlr, and the variants of genomic structure were detected by svim with "-nanopore" option [42, 43]. Then, the structural variants of BND of *torimochi* were extracted and the 36 inserts that have reliable junctions at at least one end of transposon and inside of *torimochi* were identified. The results were visualized using the circlize package in R. A representative insertion was confirmed by PCR (S1 Table).

## Adipocyte differentiation of BmN4 cells

For the induction of adipocyte differentiation, BmN4 cells were cultured in IPL-41 medium supplemented with 10% fetal bovine serum, 10 μg/ml insulin (SIGMA), 0.5 μM 3-isobutyl-1-methylxanthine (SIGMA) and 250 μM dexamethasone (SIGMA). Medium was changed twice a week and collected 10 and 20 days after the drug addition to prepare RNA sequence libraries. These libraries were mapped to the silkworm genome and gene-model, transposon, and the novel inserts identified in this paper using Hisat2 with the default parameters, and the mapped reads were calculated by coverageBed of bedtools [41, 49]. Silkworm homologs of piRNA-related genes were identified using blast of silk-base. Since the genemodel for BmAgo3 was splited into three (KWMTBOMO01223-5) and the genemodel for Tudor was splited into two (KWMTBOMO06482-3), the RPKM was calculated by combining them.

## Comparison of *torimochi* insertion sites in different lines of BmN4 cells

The genome sequence of BmY was also sequenced by MinION in the same manner as described above, and the insert position of *torimochi* was identified by svim. Sequences with and without *torimochi* inserted at these insert positions were prepared using R scripts. Min-ION sequencing data for BmQ and BmY were mapped to these sequences using bwa-mem with "ont2d" option. The presence or absence of the reads that cover at least 100 bps at both sides of a given junction site was used as a criterion.

## Supporting information

**S1 Fig. Basic characteristics of *torimochi* and *torimochi*-derived piRNAs.** (A) The length distribution of *torimochi*-derived small RNAs in BmN4 cells. (B) Schematics for the ping-pong signature (C) and the phasing signature (D). (C) The fraction of 5'-5' overlapped piRNAs (y-axis) at the indicated length (x-axis) for *torimochi*-derived piRNAs in the small RNA libraries from BmN4 cells. The Z-scores at the 10 nt overlap (ping-pong signature) are shown in the top-right of the graphs. (D) The fraction of 3'-5' distance for *torimochi*-derived piRNAs in naive BmN4 cells or BmN4 cells knocked out for Trimmer, the exonuclease that trims the 3'-end of precursor piRNAs for maturation (TriKO). $Z_1$ denotes the z score at position 1. A strong tail-to-head phasing signature (at position = 1) was detected in *torimochi*-derived piRNAs from TriKO cells. (E) In the chr12 region of the p50T genome, *torimochi* is nested within another transposon, LINE/R1. This LINE/R1 element is extremely GC-rich and repetitive, which likely hinders the PCR amplification of the full-length sequence in Fig 1F. On the other hand, a strong band was detected at ~2.0 kb for BmN4 (Fig 1F), which indicates the absence of both torimochi and LINE/R1 in this region.
(TIFF)

**S2 Fig. The novel *torimochi* inserts are specific to BmN4 cells.** (A–I) Genomic PCR of the *torimochi* inserts newly identified in BmN4 cells. The genome DNAs of BmN4 cells and silk-worm strains p50T, N4, and C108T and *B. mandarina* (*Bma*) were used. Black and white arrows indicate the band lengths with and without *torimochi*, respectively. (G) The genome of the N4 strain has a *torimochi* insertion at the same site.
(TIFF)

**S3 Fig. piRNA production inside and outside of the newly identified *torimochi* sequences.** (A–H) Distribution of *torimochi*-derived piRNAs in BmN4 cells and ovaries. The boundaries of *torimochi* are shown by gray lines. (I) Schematic diagram of *torimochi*, GFP transgene, and the outside genomic regions on chromosome 13. The MinION reads that span these regions

are shown by black lines. (J) piRNA production from the *torimochi* copy on chromosome 13 in BmN4 cells and ovaries
(TIFF)

**S4 Fig. Representative sequences of novel transpositions in the BmN4 genome.** (A–L) Domain structure of the representative sequence of each group (top), the region where it is originally annotated in the p50T genome (left), and the lengths of the inserts found in BmN4 cells (bottom). ORF: ORFs that could not be annotated by Pfam. Names of newly identified transposons are indicated in red.
(TIFF)

**S5 Fig. Many endogenous transposons are inserted into *torimochi* in BmN4 cells.** (A) Transposons inserted in *torimochi* and their locations. Transposons with at least one end (within 100 bp) forming a junction with a sequence inside *torimochi* were considered to be inserted into *torimochi*. Seven out of the 36 inserts accounted for *mejiro* (S4A Fig). The red line shows the insertion whose presence was confirmed by genomic PCR in S5B and S5C Fig. (B) The insertion of mejiro within *torimochi* shown as red in S5A Fig. The primer sets used in C are shown by black bars. (C) Genomic PCR shows the transposon insertion specifically in BmN4 cells.
(TIFF)

**S6 Fig. Comparison of *torimochi* insertion sites in different lines of BmN4 cells.** (A) Definition of *torimochi* insertion. The presence or absence of the reads that cover at least 100 bps at both sides of a given junction site was used as a criterion in (B). (B) The *torimochi* insertion sites identified in the BmQ (left) or BmY (right) genome. The ratio of MinION reads from BmQ (top) or BmY (bottom) cells with or without *torimochi* sequence was plotted for each identified insertion site.
(TIFF)

**S1 Table. Primers used in this study.**
(PDF)

**S2 Table. Annotation of representative sequences for each group.** The longest sequence in each group as representative sequence was annotated by BLASTn. Green or gray rows are those that could not be illustrated in S3 Fig due to many variations in the sequence. Green group's representative sequences are LINE/SINE and gray are others.
(PDF)

**S3 Table. Regions with homologous sequences to the newly identified insert groups.** The "old_area" represents the length of homologous regions in the p50T genome, and the "new_-area" represents the total length of the inserts identified in this study.
(PDF)

## Acknowledgments

We thank Yutaka Suzuki for sequencing for differentiated BmN4 cells, Takashi Kiuchi for providing silkworm genomic DNA and Kaori Kiyokawa for technical assistance. We thank all the members of the Tomari laboratory for discussion and critical comments on the manuscript.

## Author Contributions

**Conceptualization:** Keisuke Shoji, Yukihide Tomari.

**Data curation:** Keisuke Shoji, Yusuke Umemura.

**Formal analysis:** Keisuke Shoji, Yusuke Umemura.

**Funding acquisition:** Keisuke Shoji, Susumu Katsuma, Yukihide Tomari.

**Investigation:** Keisuke Shoji.

**Methodology:** Keisuke Shoji, Susumu Katsuma.

**Project administration:** Keisuke Shoji, Yukihide Tomari.

**Supervision:** Susumu Katsuma, Yukihide Tomari.

**Validation:** Keisuke Shoji, Yusuke Umemura.

**Visualization:** Keisuke Shoji.

**Writing – original draft:** Keisuke Shoji.

**Writing – review & editing:** Yukihide Tomari.

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
