## [Editor Report · Decision Letter 0]

21 Oct 2022

Dear Dr Shoji,

Thank you very much for submitting your Research Article entitled 'The piRNA cluster torimochi is an expanding transposon in cultured silkworm cells' to PLOS Genetics.

The manuscript was fully evaluated at the editorial level and by independent peer reviewers. The reviewers appreciated the attention to an important problem, but raised some substantial concerns about the current manuscript. Based on the reviews, we will not be able to accept this version of the manuscript, but we would be willing to review a much-revised version. We cannot, of course, promise publication at that time.

If you decide to revise the manuscript for further consideration at PLOS Genetics, please aim to resubmit within the next 60 days, unless it will take extra time to address the concerns of the reviewers, in which case we would appreciate an expected resubmission date by email to plosgenetics@plos.org.

We are sorry that we cannot be more positive about your manuscript at this stage. Please do not hesitate to contact us if you have any concerns or questions.

Yours sincerely,

René F. Ketting

Guest Editor

PLOS Genetics

Wendy Bickmore

Section Editor

PLOS Genetics

---

## [Decision Letter · Decision Letter 1]

20 Jan 2023

Dear Dr Shoji,

We are pleased to inform you that your manuscript entitled "The piRNA cluster torimochi is an expanding transposon in cultured silkworm cells" has been editorially accepted for publication in PLOS Genetics. Congratulations!

Yours sincerely,

René F. Ketting

Guest Editor

PLOS Genetics

Wendy Bickmore

Section Editor

PLOS Genetics

Comments from the reviewers (if applicable):

Reviewer's Responses to Questions

**Comments to the Authors:**

Reviewer #1: The authors of done a very good job of addressing the concerns raised in the first submission. The findings are intriguing and may shed light on piRNA cluster evolution. the finding will be of significant interest to small silencing RNA and transposon communities.

**Have all data underlying the figures and results presented in the manuscript been provided?**

Reviewer #1: Yes

PLOS authors have the option to publish the peer review history of their article (what does this mean?). If published, this will include your full peer review and any attached files.

Reviewer #1: No

**Data Deposition**

http://datadryad.org/submit?journalID=pgenetics&manu=PGENETICS-D-22-01104R1

**Press Queries**

---

## [Editor Report · Acceptance letter]

3 Feb 2023

PGENETICS-D-22-01104R1 

The piRNA cluster torimochi is an expanding transposon in cultured silkworm cells 

Dear Dr Shoji, 

We are pleased to inform you that your manuscript entitled "The piRNA cluster torimochi is an expanding transposon in cultured silkworm cells" has been formally accepted for publication in PLOS Genetics! Your manuscript is now with our production department and you will be notified of the publication date in due course.

With kind regards,

Judit Kozma

PLOS Genetics

On behalf of:
